# Untangling the Complexity of Two-Component Signal Transduction in Bacteria

**DOI:** 10.3390/microorganisms13092013

**Published:** 2025-08-28

**Authors:** Patrycja Wadach, Dagmara Jakimowicz, Martyna Gongerowska-Jac

**Affiliations:** Department of Molecular Microbiology, Faculty of Biotechnology, University of Wroclaw, 50-383 Wroclaw, Poland; patrycja.wadach2@uwr.edu.pl (P.W.); dagmara.jakimowicz@uwr.edu.pl (D.J.)

**Keywords:** two-component system, bacteria, gene transcription regulation, histidine kinase, response regulator, phosphorylation, phosphorelay

## Abstract

Two-component systems (TCSs) are ubiquitous in bacteria and are central to their ability to sense and respond to diverse environmental and intracellular cues. Classically composed of a sensor histidine kinase and a cognate response regulator, TCSs control processes ranging from metabolism and development to virulence and antibiotic resistance. In addition to their biological roles, TCSs are garnering attention in synthetic biology and antimicrobial drug development. While canonical architectures have been extensively studied, increasing evidence highlights the remarkable diversity in their organization and regulation. Despite substantial progress, key questions remain regarding the prevalence and physiological relevance of non-canonical TCSs, the mechanisms ensuring signal fidelity, and the potential for engineering these systems. This review explores non-typical TCSs, focusing on their varied transcriptional regulation, alternative response regulator activities, varied control by phosphorylation, and negative control mechanisms. We discuss how bacteria manage signaling specificity among numerous TCSs through cross-talk, hierarchical interactions, and phosphorelay systems and how these features shape adaptive responses. By synthesizing current understanding and highlighting still existing knowledge gaps, this review offers a novel perspective on TCS diversity, indicating directions for future research and potential translational applications in biotechnology and medicine.

## 1. Introduction

In all domains of life, gene expression is precisely controlled, primarily at the transcription initiation level. Specific cellular processes necessitate the integration of various signals to determine whether to increase or decrease the production of a particular gene product. Crucial elements in this multistep control are the *cis*-acting elements in the gene promoter sequence, as well as *trans*-acting proteins known as transcriptional regulators. Transcriptional regulators act as repressors or activators by affecting RNA polymerase (RNAP) binding to DNA. The binding of transcription factors allows gene expression to be adjusted to changes in environmental conditions. To this end, their binding to DNA is meticulously regulated by numerous factors, including external and internal signals, through a plethora of mechanisms [1]. These mechanisms include posttranslational modifications [2], ligand binding [3,4], protein partner interactions/oligomerization [5], and proteolysis [6], as well as the accessibility of the recognized sequences, which in turn may be regulated by DNA topology [7,8].

In bacteria, a particular group of transcription factors belong to two-component regulatory systems (TCSs). In these systems, the activity of transcriptional regulators is under the control of dedicated kinases. In addition to bacteria, they are also present in archaea and eukaryotes (but absent from the animal kingdom) [9]. Each bacterial species possesses a range of TCSs, typically from a few tens to a few hundreds of systems. These transduction systems serve to couple the detection of a signal with an appropriate cellular response. Certain TCSs play crucial roles in promoting growth and improving cellular conditions by adjusting metabolism, whereas others are linked to triggering virulence and the development of drug resistance or tolerance [10]. Consequently, targeting these TCSs is suggested as a feasible approach for antimicrobial therapy [11]. Thus, understanding the mechanism of TCS action is essential for advancing the field.

Many TCSs have been thoroughly studied. Among them is the PhoBR system (or homologous PhoPR, named depending on the genus) involved in controlling phosphate metabolism, which was first characterized in *Escherichia coli* and later characterized in many other bacterial species [9]. Similarly, many investigations have focused on EnvZ/OmpR, which is involved in osmoregulation [12]; NtrBC, which is a key regulator of nitrogen assimilation [13]; NarXL, which is involved in nitrate regulation [14]; and the actinobacterial MtrAB, which is involved in cell wall metabolism [15]. In recent years, several reviews have addressed bacterial two-component systems (TCSs). Most of these, however, focus either on specific bacterial species [16,17,18,19,20,21] or on individual TCSs [12,22,23]. Broader surveys that provide a comprehensive overview of TCS diversity were mostly published two decades ago [24,25,26,27,28], while more recent reviews typically concentrate on particular aspects, such as cross-talk, histidine kinase structure and function, phosphorylation mechanisms, or the roles of TCSs in antibiotic resistance, virulence, and biotechnological applications [29,30,31,32,33,34].

Throughout the last decades, the unexpected complexity of their mechanisms of action has been revealed. Additionally, despite the significant progress made in understanding the structure and function of individual systems, additional aspects have attracted considerable attention. How do bacterial cells coordinate many similar signaling pathways? Even though the kinases and regulators within an organism are often very similar, cells can still match specific inputs to the correct outputs. How do they avoid unwanted cross-talk? Do they use the similarities between these proteins to integrate signals or diversify their responses? A related area of interest is understanding the evolution of these systems. How do two-component pathways evolve, and how are new pathways introduced? Furthermore, how are new pathways distinct from each other?

In our view, there was a need for a review emphasizing the diversity of TCSs, their functions, and their unique features across different bacterial species. In this review, we therefore aim to present the current state of knowledge on non-typical TCSs, discussing their organization, activities, and modes of regulation. The presentation of a set of examples of uncommon TCSs is intended to reveal an extremely wide repertoire of their functions and mechanisms of action.

## 2. Canonical TCS

The prototypical TCS comprises a sensor histidine kinase (HK) and a response regulator (RR). In typical TCSs, HKs act as donors of phosphate groups to their respective RRs, leading to the correct output responses. The HK is frequently (but not always) a membrane-associated protein that shares fundamental signaling components: a diverse sensing domain, a signal-transducing domain (also known as the transmitter domain or linker region), and a highly conserved kinase core composed of a dimerization and histidine phosphotransfer (DHp) domain, as well as an ATP/ADP-binding phosphotransfer catalytic (CA) domain (Figure 1A).

The N-terminal sensing domains of HKs detect environmental stimuli directly or indirectly. In transmembrane HKs, the sensing domain is connected to the cytoplasmic kinase core domain through a transmembrane helix and a cytoplasmic linker of variable lengths. Although the transmitter domain remains the least understood segment of any HK, it is critical for proper signal transduction, as indicated by multiple studies [24,35,36,37,38].

Despite many HKs being membrane associated, they might respond to signals within the cytoplasm, such as changes in ion concentrations, pH, ligand availability and redox state [39,40]. While most HKs are located within the membrane, soluble HKs are also regulated by intracellular signals or through interactions with the cytoplasmic domains of other proteins [40,41,42]. For example, FixL from *Bradyrhizobium japonicum* senses oxygen concentration in the cytoplasm, while ShkA from *Caulobacter crescentus* detects the level of the secondary messenger c-di-GMP [43,44]. These cases highlight how canonical HK–RR systems can be wired to intracellular signals, foreshadowing the broader integration of RRs into complex regulatory networks.

Transmembrane and membrane-anchored HKs can also respond to signals originating from the membrane itself or from within the cytoplasm. The *B. subtilis* HK DesK lacks extracellular domains and instead relies on changes in membrane thickness or composition to sense temperature fluctuations, thereby linking cytoplasmic conditions to cellular responses [45]. Recent findings have also shown that DesK is capable of sensing cytoplasmic pH changes through its linker domain, located between the transmembrane and DHp domains [46]. DesK exemplifies how canonical systems adapt sensing modules for multiple inputs, an adaptability mirrored in the functional diversity of their partner RRs. This finding suggests a broader functional versality in how these proteins sense and respond to cellular environments [47].

The cytoplasmic part of HKs consists of a signal-transducing domain and a kinase core domain that is highly conserved and defines HK function (Figure 1A). The intracellular signal-transducing domain(s) contains domains that help transmit the signal to the catalytic core of the kinase domain. These may be HAMP (present in histidine kinases, adenylate cyclases, methyl-accepting proteins, and phosphatases), PAS (Per-Arnt-Sim), or GAF (cGMP-specific phosphodiesterases, adenylyl cyclases, and FhlA) domains that are specialized to detect different stimuli. These transducing domains often exist as combinations or tandem repeats and are found in many other bacterial and mammalian proteins [47].

The kinase core domain consists of a DHp domain and CA domain (Figure 1A). The DHp domain contains a conserved histidine residue that undergoes autophosphorylation. This domain is responsible for the dimerization of HK. The DHp transfers a phosphate to the RR. The CA domain binds ATP to transfer a phosphate to the histidine in the DHp domain [47,48].

RRs consist of a conserved, regulatory N-terminal receiver domain and a highly variable C-terminal output (effector) domain separated by a flexible linker (Figure 1B). The receiver (REC) domain contains aspartate residues (phosphoacceptors) adjacent to other acidic residues. They are involved in the coordination of the Mg^2+^ ion that is required for phosphoryl transfer and dephosphorylation [24]. Phosphorylation induces conformational changes that activate the effector domain.

The effector domain determines the specific functions of RRs. Typically, the effector domain contains a helix-turn-helix (HTH) DNA-binding motif (as in OmpR, PhoB, and NarL); phosphorylation of these RRs promotes their dimerization, enhancing DNA binding. However, the effector domains of some RRs may have other properties than DNA binding (see below).

After a signal is detected, the sensor HK undergoes autophosphorylation at a conserved histidine residue within the cytoplasmic kinase core domain [24]. Once activated, a phosphoryl group is translocated from the HK to a conserved aspartate residue in the N-terminal REC domain of a cognate RR (Figure 2). Unlike typical protein kinase cascades, where one kinase phosphorylates multiple targets, TCSs operate through the stoichiometric transfer of the phosphoryl group to a single receiver [24]. In addition to driving the forward phosphorylation reaction, many HKs also exhibit phosphatase activity, allowing them to dephosphorylate their cognate RRs. Moreover, the RR activity may be regulated not only by HK, but also other mechanisms (see below).

## 3. Transcriptional Regulation of TCS Genes

Most genes encoding proteins in TCSs are transcriptionally linked, i.e., encoded in operons, and controlled by the same promoter. This gene arrangement ensures that the sensor controls the phosphorylation of its specific regulator. In *E. coli*, approximately 50% of TCS-encoding genes are organized with the RR gene upstream of the kinase gene, often leading to higher levels of the regulator than the sensor kinase (e.g., the *ompR*/*envZ* system [12]). However, even among homologous TCS genes, the expression patterns can vary across bacterial species (Table 1).

A key feature of many TCS-encoding genes is positive autoregulation, where the phosphorylated RR increases the expression of genes encoding both the corresponding HK and RR proteins. Higher levels of these proteins can increase the sensitivity of the TCS to external stimuli and strengthen the adaptive response. When the external stimulus is strong and long-lasting, positive autoregulation intensifies the adaptive reaction. Conversely, if the stimulus is weak or transient, autoregulation can lead to an amplified or prolonged response relative to the actual stimulus, which may be considered inappropriate under certain conditions [49]. Autoregulation usually allows the system to adjust gene expression finely, enabling different levels of the RR to control distinct sets of genes. For example, in *Salmonella*, low levels of PhoP still affect the expression of some target genes while eliminating its impact on other genes of its regulon [26]. Autoregulation creates a hierarchy of gene expression based on the number of active regulator molecules and the properties of target promoters. In the PhoB/PhoR system of *E. coli*, for example, genes involved in phosphate uptake are activated first, followed by phosphate-scavenging genes [50]. Autoregulation also helps bacteria “remember” signals, maintaining higher protein levels for faster responses upon re-exposure, as observed in the PhoBR system [51]. This mechanism is important for systems in which the RR controls numerous target sites with limited amounts of protein molecules (e.g., PhoQ/PhoP and CpxA/CpxR). In some systems, autoregulation sets a threshold, ensuring that only persistent signals lead to gene activation, as observed in the BvgA/BvgS system of *Bordetella pertussis* [52,53].

Many TCS genes are controlled by both a constitutive promoter for baseline expression and an autoregulated promoter to increase regulator levels in response to environmental signals. Typically, TCS genes must be transcribed at a low level to produce sufficient sensor and regulator proteins for signal detection and response. However, exceptions exist. For example, in *Streptomyces coelicolor*, the SitKR system is not transcribed under optimal conditions and only becomes active under specific conditions, such as high DNA supercoiling or through another TCS [54]. In some TCS operons, only a single promoter provides both basal and autoregulated expression (e.g., the ComDE system in *Streptococcus pneumoniae*, [55]). However, the RR gene often has its own additional promoter sequence. Interestingly, some TCS proteins are encoded by genes that are transcribed independently and may be located in distant genome regions [27]. For example, in *Salmonella*, the transcription of two components, SsrA and SsrB, is not coupled, meaning that SsrB can be produced without SsrA and vice versa [56].

The regulation of TCS protein levels serves multiple functions, depending on the system, and new regulatory interactions are continuously being discovered through transcriptional and genetic studies, which may eventually allow predictions of bacterial behaviors in different environments. The diversity of transcriptional organizations—from tightly coupled operons with shared promoters, through autoregulated feedback loops, to systems with independently transcribed or conditionally activated components—illustrates that bacteria have evolved multiple strategies to fine-tune TCS activity. Operon-based arrangements preserve stoichiometry and ensure synchronized production of HKs and RRs, whereas independent promoters or uncoupled transcription allow greater flexibility, differential protein abundance, and rewiring of regulatory networks. Such variability not only reflects evolutionary pressures to adapt signaling architectures to ecological niches but also provides a conceptual toolkit for synthetic biology. By exploiting these natural design principles, TCSs can be re-engineered for predictable signal processing, tunable sensitivity, and programmable regulatory hierarchies in engineered microbial systems.

**Table 1 microorganisms-13-02013-t001:** Transcriptional organization of genes encoding TCS proteins in different bacterial species.

HK/RR	Organism	Function Description	Transcriptional Organization	References
EnvZ/OmpR	*E. coli*	Osmoregulation	Classical operon; background transcription, upregulated under external osmotic stimuli	[12]
PhoR/PhoB	*E. coli*	Phosphate metabolism	Classical operon; transcription is constitutive and positively autoregulated under phosphate-limiting conditions	[50,51]
CpxA/CpxR	*E. coli*	Envelope stress response system	Classical operon; transcription is constitutive and positively autoregulated under envelope stress	[57,58]
BvgS/BvgA	*B. pertussis*	Virulence gene regulation	Classical operon; transcription is constitutive and positively autoregulated under virulence-promoting conditions	[52,53,59]
ComD/ComE	*S. pneumoniae*	Genetic competence	Classical operon; transcription is constitutive and positively autoregulated via competence-stimulating peptide	[55]
SitK/SitR	*S. coelicolor*	Repression of sporulation	Classical operon; transcription is conditionally activated under specific stress conditions	[54]
SsrA/SsrB	*Salmonella*	Virulence gene regulation	Nonoperonic; transcription is independent and uncoordinated from separate promoters	[56]

## 4. Variations in RR Activities

Although most RRs share sequence and structural similarities, they can utilize distinct functional effector domains [60] (Table 2). The majority of bacterial transcriptional activators specifically bind DNA sites within the regulatory regions of their target genes. This attachment serves either to recruit RNAP to the promoter or to stimulate subsequent stages of transcription initiation. Although most RRs bind DNA as dimers, exceptions exist; for example, transcriptional activation by NtrC (*Salmonella*) requires its oligomerization, and numerous NtrC molecules do not bind directly to DNA but instead activate transcription via protein-protein interactions [61,62].

Transcriptional activation may also be independent of RR binding to DNA (Table 2). For example, FlgR in *Campylobacter jejuni* was proposed to bind the σ^54^-holoenzyme directly to activate transcription independently of the C-terminal (CTD) domain [63]. A similar mechanism has been proposed for an FlgR homolog in *Helicobacter pylori*, which naturally lacks a DNA-binding CTD [64]. Moreover, RRs may not be involved in the regulation of transcription initiation but rather in transcription antitermination by binding RNA instead of DNA, like RR17 from *Enterococcus faecalis* [65], influencing translation efficiency (as in the case of AmiR [66] and NasR [67]), or controlling RNA stability (e.g., CsrA [68]). Similarly, PdtaSR from *Mycobacterium tuberculosis* is an atypical TCS with a cytosolic HK and an RR acting as a transcriptional antiterminator via its RNA-binding ANTAR (AmiR and NasR transcription antitermination regulators) domain. This domain causes the formation of an antiterminator structure to allow transcription to proceed past its normal termination point [69].

Additionally, some RR effector domains may have enzymatic activity instead of DNA/RNA affinity (Table 2). Examples include methylesterase CheB [70] and the c-di-GMP phosphodiesterase PdeR [71]. On the other hand, a group of RRs called single-domain response regulators (SDRRs) does not possess any enzymatic activity at all, as they contain only an REC domain (with a typical Asp residue being phosphorylated after activation) [72]. These proteins lack a typical output domain and induce a cellular response via protein-protein interactions, which act as molecular switches or adaptors rather than classical transcription factors. They act by directly binding enzymes, structural proteins, or scaffold complexes (e.g., *E. coli* CheY, which interacts with the flagellar motor switch protein FliM). Upon binding, they modulate the activity of kinases/phosphatases (e.g., DivK-dependent modulation of the pseudokinase DivL and the hybrid HK CckA), control the localization of protein complexes and assemble signaling platforms, and often play roles in motility, chemotaxis, and biofilm regulation (e.g., CheY) [73] (Table 2).

While the TCSs are recognized primarily as transcriptional regulators, the above examples of RR activities indicate the significant variation in their modes of action.

**Table 2 microorganisms-13-02013-t002:** Diversity of RR functions. Representative RRs and their corresponding HKs in different bacterial species. Their functional roles and mechanisms of RR activity are listed.

HK/RR	Organism	Function Description	Description of the RR Effector Domain and Its Activity	References
EnvZ/OmpR	*E. coli*	Osmoregulation	CTD DNA-binding domain, regulates gene expression	[60]
PhoR/PhoB	*E. coli*	Phosphate metabolism	CTD DNA-binding domain, regulates gene expression	[74]
SsrA/SsrB	*Salmonella* spp.	Virulence gene regulation	CTD DNA-binding domain, regulates gene expression	[56]
NtrB/NtrC	*Salmonella* spp.	Nitrogen regulation	CTD DNA-binding domain, oligomerizes to activate σ^54^-RNAP	[13,61]
FlgS/FlgR	*C. jejuni*	Flagellar gene expression	Lacks a CTD DNA-binding domain, activates transcription by interaction with σ^54^-RNAP	[63]
FlgS/FlgR	*H. pylori*	Flagellar biosynthesis	Lacks a CTD DNA-binding domain, activates transcription by interaction with σ^54^-RNAP	[64]
PdtaS/PdtaR	*M. tuberculosis*	Adaptation to stress by the regulation of RNA structure	CTD RNA-binding domain, binds nascent RNA to prevent transcriptional termination	[69]
CheA/CheB	*E. coli*	Chemotactic adaptation	CTD with enzymatic activity, the receptor methyl-esterase	[70]
CheA/CheY	*E. coli*	Flagellar motor switching	CTD involved in protein–protein interaction, binds FliM to reverse flagellar rotation	[73]
DivJ/DivK	*C. crescentus*	Controls cell cycle progression and the cell’s fate	Single domain (receiver domain), modulates DivL–CckA signaling through a protein–protein interaction	[75]
-/AmiR	*Pseudomonas aeruginosa*	Amidase operon regulation	Orphan RR, CTD RNA-binding domain, transcriptional antitermination regulator	[66,76]
-/NasR	*Klebsiella* spp.	Nitrate/nitrite assimilation	Orphan RR, CTD nitrate-activated RNA-binding domain, antiterminator	[67]
-/CsrA	*E. coli*	Global carbon metabolism, motility, biofilm formation	Orphan RR, CTD RNA-binding domain, stabilization of RNA	[68]

## 5. Regulation of RR Activity by Phosphorylation

TCSs can act either as switches, where phosphorylation fully activates or deactivates the system, or as rheostats, where varying phosphorylation levels produce a gradient of gene expression based on signal strength and duration. Researchers have suggested that RRs exist in the cell as an equilibrium mixture of distinct states, e.g., unphosphorylated, phosphorylated, unphosphorylated bound to DNA with low affinity and phosphorylated bound to DNA with high affinity [77] (Table 3). Interestingly, the binding of RRs to DNA targets can increase their autophosphorylation [78]. The phosphorylation of RRs can have different consequences and influence gene transcription in different ways. The overwhelming majority of RRs activate transcription exclusively when they are in a phosphorylated state. Canonical TCS phosphorylation of the RR promotes the binding of the effector domain to DNA, activating gene transcription; however, other modes of regulation have also been described. While in *E. coli*, OmpR phosphorylation at the N-terminus increases the DNA binding affinity of the C-terminal domain; in the closely related PhoB, the N-terminal domain prevents the C-terminal DNA-binding domain from binding to the target unless it is phosphorylated [56,60]. Phosphorylated RRs that inhibit gene expression are less often described. One example of such a negatively acting RR is ComE from *S. pneumoniae*, which acts through the repression of capsular polysaccharide genes [79]. Many RRs display a bidirectional activation mode, as they are capable of binding DNA in their unphosphorylated state, although phosphorylation increases the binding affinity and sometimes alters the specificity [80,81]. This regulation is exhibited by OmpR [82], PhoP [83], BvgA [84,85], SsrB [86], and RcsB [87]. Some RRs can act differently depending on their phosphorylation status, such as AlgR or SsrB. Phosphorylated AlgR activates processes associated with the initial stages of infection; conversely, unphosphorylated AlgR controls later stages of biofilm development by activating different sets of genes [88]. Phosphorylated SsrB activates the transcription of SPI-2; conversely, biofilm-related gene transcription is activated by SsrB independently of its phosphorylation [86,89]. Another well-studied example is DegU from *B. subtilis*, which, when phosphorylated, regulates motility genes [90,91,92], and the unphosphorylated form controls competence-related genes [93]. Additionally, in FimZ from *E. coli*, one activity (induction of pili gene transcription) is dependent on phosphorylation, but the second activity (sustained interaction with the α subunit of the F_1_ ATP synthase determining cell elongation) is independent of phosphorylation [94].

Interestingly, phosphorylation may not control the DNA-binding affinity of the RR but instead promotes the formation of an oligomeric form of the protein (e.g., ComE from *Staphylococcus*) [95]. Some RRs can oligomerize and bind to DNA only in the dephosphorylated state, while phosphorylation causes their dissociation from DNA and derepression of transcription. One example of such a protein is HnoC from *Shewanella oneidensis*, which is a nitric oxide RR that controls biofilm formation. Unphosphorylated HnoC exists as a tetramer and binds tightly to DNA, whereas phosphorylation causes subunit dissociation and transcriptional derepression [96]. The phosphorylation of RRs may not directly regulate their DNA binding but rather release the inhibition of transcription, which is governed by other factors (e.g., OmpR from *Salmonella* releases H-NS inhibition under acid stress) [97].

Although phosphorylation predominantly modifies the DNA affinity of the C-terminal domain of RR, the interdomain linker is also important for target gene recognition and the regulation of RR activity [60]. Moreover, RR binding to DNA may be controlled not only by phosphorylation but also by another protein. An example is SwrA, a small coactivator protein that increases DegU affinity for DNA and enhances its binding to the target promoter, likely by inducing its oligomerization. DegU oligomerization induced by SwrA is necessary to allow both the remote binding of DegU and interaction with RNAP at the promoter. The combined action of SwrA and DegU is critical for initiating the process of building flagella [92]. Recent studies have also shown that *B. subtilis* DesR, as well as E. coli OmpR, can maintain their active conformations not only through phosphorylation, but also via protein–protein interactions with their cognate HKs, DesK, and EnvZ, respectively [98].

The overview of RR regulation strategies presented above clearly shows the complexity of the systems and explains the mechanisms underlying the regulatory network that involves the TCS.

## 6. Nonspecific Phosphorylation and Acetylation

The complexity of TCS regulation also increases because cognate HKs are not the only molecules that can phosphorylate RRs (Table 3). Acetyl phosphate (AcP) may act as a phosphate donor for numerous RRs without involving sensor kinases or ATP [99]. AcP is synthesized from acetyl-CoA and phosphate by a phosphotransacetylase encoded by *pta* or from ATP and acetate by an acetate kinase encoded by *ackA*. Some RRs can be efficiently phosphorylated by both cognate HKs and AcP (e.g., DegU from *B. subtilis* [92,100,101]). This phosphorylation is particularly relevant in the absence of a cognate HK or when the kinase is inactive. However, some RRs may have evolved HK-independent mechanisms to prevent cross-talk with AcP or other small-molecule phosphodonors. For instance, C-terminal domain of FlgR from *C. jejuni* is a specificity determinant to limit in vivo cross-talk from AcP [64]. In some other instances, such as PhoP in *B. subtilis* or CopR from *P. aeruginosa*, RRs cannot undergo phosphorylation with AcP without kinase activity [83,102].

**Table 3 microorganisms-13-02013-t003:** Functional diversity of RRs driven by an activation mechanism. Variations in the dependence of RRs on phosphorylation and alternative regulatory inputs (e.g., acetylation or dual roles).

RR Name	Organism	Biological Function	Regulation by Phosphorylation	References
ComE	*S. pneumoniae*	Capsule assembly	HK-dependent; gene repression by phosphorylated RR	[79]
OmpR	*E. coli*	Osmoregulation	HK- or AcP-dependent; gene activation in both states: binds DNA when unphosphorylated and phosphorylation increases the specificity	[82]
PhoP	*B. subtilis*	Phosphate metabolism	HK-dependent; gene activation in both states: binds DNA when unphosphorylated and phosphorylation increases the specificity	[83]
BvgA	*B. pertussis*	Regulation of virulence gene expression	HK-dependent; gene activation in both states: binds DNA when unphosphorylated and phosphorylation increases the specificity	[84,85]
RcsB	*E. coli*	Envelope stress; capsule regulation	HK- or AcP-dependent; gene activation in both states: binds DNA when unphosphorylated and phosphorylation increases the specificity	[103]
SsrB	*Salmonella* spp.	SPI-2 transcription; biofilm formation	HK-dependent; stage-dependent phosphorylated and nonphosphorylated forms activate different sets of genes	[86]
AlgR	*P. aeruginosa*	Infection; biofilm formation	HK-dependent; stage-dependent phosphorylated and nonphosphorylated forms activate different sets of genes	[88]
DegU	*B. subtilis*	Motility; competence	HK- or AcP-dependent; phosphorylated and nonphosphorylated forms activate different sets of genes	[92,100,101]
FimZ	*E. coli*	Pili expression; pili elongation	Orphan RR; unknown phosphodonor; phosphorylated and nonphosphorylated forms activate different sets of genes	[94]
HnoC	*S. oneidensis*	Biofilm formation in response to nitric oxide	HK-dependent; the unphosphorylated protein binds as tetramer and represses transcription	[96]
Rrp2	*B. burgdorferi*	Virulence	Orphan RR; acetyl phosphate-dependent activation	[64]
CheY	*E. coli*	Chemotactic motor switching	HK- or AcP-dependent; activated by acetylation	[104]
TcrX	*M. tuberculosis*	Acid-sensing persistence regulon	HK-dependent; acetylation modulates phosphorylation	[105]
MtrA	*M. tuberculosis*	Cell cycle/cell wall gene regulation	HK-dependent; acetylation modulates phosphorylation	[105]

While phosphorylation is a typical modification that changes TCS activity, the acetylation of RRs can also affect their DNA binding and transcriptional regulatory activities (Table 3). Lysine residues of RRs can be acetylated either catalytically with acetyl-CoA by acetyltransferase (encoded by *yfiQ* in *E. coli*) or nonenzymatically by AcP [106]. Deacetylases (encoded by *cobB* in *E. coli*) may also control the level of the acetylated RR, as is the case for RcsB in *E. coli* [106,107]. The acetylation of RcsB inhibits its activity [103]. In contrast, the in vitro acetylation of *E. coli* CheY was shown to activate it through a mechanism that is not yet fully understood [104,108]. Interestingly, some studies have revealed a unique signaling system in *M. tuberculosis* in which acetylation increases the phosphorylation of RR TcrX by cognate kinases while reducing its phosphorylation by noncognate kinases (such as MtrB). Moreover, another RR from this organism, MtrA, is similarly affected by acetylation, resulting in increased phosphorylation by its cognate kinase, MtrB [103]. Thus, acetylation contributes to controlling phosphorylation specificity; however, nonenzymatic and nonspecific acetylation by acP is a more global phenomenon than the enzymatic acetylation of RRs.

## 7. Orphan RRs

Orphan RRs are structurally typical RRs; however, their cognate (genomically linked) HKs are unknown. Although paired HKs and RRs function efficiently, orphan RRs are universal in bacteria. For example, 2 of the 20 RRs in *Streptococcus sobrinus* and 2 of the 16 RRs in *Acinetobacter baumannii* are orphan RRs. In comparison, *Myxococcus xanthus* has a total of 119 RRs, more than 50% of which are orphans; however, in *E. coli*, only 1 (FimZ) of the 34 RRs is an orphan.

How orphan RRs become phosphorylated remains unclear (Figure 3). Hypotheses include cross-talk from noncognate HKs or phosphorylation by small molecules (e.g., acetyl phosphate, see above). However, orphan RRs often do not rely on phosphorylation for activation, and in fact, the ability to regulate gene expression independently of phosphorylation (e.g., via protein-protein interactions) is a typical feature of orphan RRs (see also Table 3) [81].

## 8. Role of Serine-Threonine Kinases in RR Phosphorylation

In prokaryotes, His-Asp phosphorelay systems of TCSs are believed to be the primary regulatory systems that respond to and translate environmental cues into cellular responses. Nevertheless, STKs may also contribute to signal transduction by the TCS (Table 4). Recent studies have shown that these two pathways may overlap: both kinases may control the same regulators but may sense different aspects or act under different conditions. Usually, STK-dependent phosphorylation acts as a second activator for the TCS system to increase RR activity even when HK activity decreases. Thr phosphorylation and Asp phosphorylation can cooperatively enhance RR binding to DNA targets, which indicates that STK-driven phosphorylation may be required for the full induction of some TCS regulons (such as DosR in *M. tuberculosis*) [109]. Some RRs (e.g., RitR from *S. pneumoniae* and *Staphylococcus aureus* GraR and VraR proteins), in addition to HK-driven modifications, are phosphorylated by STKs in the DNA-binding domain, suggesting an impact on the affinity of the RR for its DNA-binding sites. Alternatively, RRs can be phosphorylated in the REC domain (e.g., streptococcal CovR, mycobacterial DosR, *B. subtilis* WalR, and staphylococcal VraR), and this modification was suggested to influence protein dimerization, which is crucial for DNA binding (Figure 4) [110,111,112]. Moreover, recent studies have shown that the phosphorylation of RRs by STKs can also affect Asp phosphorylation by a cognate HK (in the case of CovR) and thus inhibit the DNA binding of RR [111,113]. In addition to the involvement of separate STKs in regulating two-component system pathways, *E. coli* HK KdpD was demonstrated to possess an additional serine kinase domain, forming a so-called tandem kinase [114]. This arrangement, associated with a Walker motif previously annotated as an ATPase, has also been identified in other HKs [115,116] and suggests that unrecognized kinase activities may be more widespread than previously anticipated, thereby broadening the repertoire of bacterial phosphorylation strategies. The involvement of secondary phosphorylation by STKs demonstrates the convergence of the two major signal transduction systems.

As STKs can directly influence TCS-regulated processes, their inhibition may suppress RR-driven cellular responses. Inhibitors of STKs have therefore attracted attention as potential antimicrobial agents, and several promising candidates have been reported in recent studies [117,118]. Beyond drug development, phosphorylation events mediated by STKs—and their ability to interface with TCS pathways—also provide a compelling basis for the design of synthetic biosensors. In such systems, phosphorylation can function as a switchable node, enabling engineered circuits to translate specific stimuli (e.g., metabolites or environmental cues) into measurable outputs. Although practical implementations of STK–TCS hybrid biosensors have yet to be investigated, the concept is robust and ripe for exploration, representing a promising new frontier in biosensing applications [119,120].

**Table 4 microorganisms-13-02013-t004:** Examples of TCSs dependent on STKs. Examples of RRs dependent on STK-mediated phosphorylation are listed.

RR Name	Organism	Biological Functions	STK Target Site in the RR	Effect of STK Phosphorylation	References
DosR	*M. tuberculosis*	Hypoxia response; latency regulon control	REC domain	Enhances DNA binding affinity	[109]
GraR	*S. aureus*	Resistance to antimicrobial peptides	DNA-binding domain	Enhances DNA binding affinity	[113,121]
VraR	*S. aureus*	Cell wall stress response and antibiotic resistance	DNA-binding domain and REC domain	Enhances DNA binding affinity	[112,113]
RitR	*S. pneumoniae*	Iron uptake and oxidative stress response	DNA-binding domain	Enhances DNA binding affinity	[113,122]
CovR	*Streptococcus* spp.	Regulation of virulence genes; immune evasion	REC domain	Inhibits Asp phosphorylation and DNA binding	[111]
WalR	*B. subtilis*	Cell wall homeostasis; control of peptidoglycan synthesis	REC domain	Modulates dimerization and DNA binding	[112]

## 9. Negative Control of TCSs

Mechanisms of negative control are responsible for preventing overactivation to maintain precise signal regulation and prevent excessive or inappropriate responses that may be detrimental to the cell. Negative control also helps fine-tune response sensitivity, reducing noise and improving accuracy. The deactivation mechanism relies mainly on the phosphatase activity of the sensor kinase (or, optionally, other cellular phosphatases) but also on proteolytic degradation or transcriptional regulation. These processes may involve the actions of accessory proteins or small RNAs and a negative feedback loop (regulation of its own transcription or transcription of their negative regulators) [27] (Table 5).

The most important and efficient inactivation of TCS proteins is their dephosphorylation. The phosphoreceiver domains of HKs naturally dephosphorylate at varying rates, with half-lives ranging from seconds to hours. However, in many TCSs, the transmitter domain significantly accelerates this process, promoting HK deactivation. This activity is known as transmitter phosphatase activity. This intrinsic autophosphatase activity helps CheY-P rapidly lose its phosphoryl group (half-life of 15 s) (Table 5) [107]. RR can also be dephosphorylated; however, spontaneous RR dephosphorylation is less probable, as it lasts too long for the signal to be deactivated. Thus, the process is regulated by cognate sensor kinases, which are often bifunctional (such as EnvZ in *E. coli*): they have both kinase and phosphatase activity, and in the absence of signals, they dephosphorylate the RR, keeping it inactive [27]. HKs are connected to their N-terminal sensor modules by coiled-coil linkers, which regulate the activity of the kinase/phosphatase and undergo signal-dependent conformational changes that underlie the K⇄P switch [123]. Certain HKs might lack one of these two functions, as observed for the bacteriophytochrome (BphP) from *Deinococcus radiodurans*, which lacks kinase activity and functions solely as a red-light-activated phosphatase [123,124]. Similarly, in *P. aeruginosa*, CopS HK activity depends mostly on its phosphatase activity, and without a signal (Cu^+^), HK maintains the cognate RR in the dephosphorylated state. After its phosphatase activity is inhibited upon Cu^+^ binding, the cognate RR is phosphorylated [102]. In many cases, separate phosphatases are also engaged in the control of TCS activity (Table 5; see also the negative control in the Phosphorelay Section).

The negative control of TCS activity in bacteria is frequently mediated by diverse, structurally unrelated proteins that modulate HKs. These proteins can either inhibit the kinase activity of HK (like Sda and KipI, which inhibit KinA in *B. subtilis*, and MgrB, which inhibits PhoQ in *E. coli*) [125,126,127]; stimulate phosphatase activity (like the PII protein, which switches NtrB to a phosphatase state in *E. coli*) [128]; or directly interact with HKs to modulate the kinase/phosphatase balance (like MzrA in *E. coli*) [129]. The direct regulation of RRs is most common in complex phosphorelay pathways, where proteins other than HKs can also directly influence the level of phosphorylated RRs (such as DivJ kinase and PleC phosphatase, which control DivK phosphorylation in *C. crescentus* [130] and the sporulation-associated cascade in *B. subtilis* [131]).

Sometimes, RRs can also affect downstream HK activity/levels. FixT, an SDRR, reportedly acts as a negative regulator of the HK FixL in *R. meliloti* and *C. crescentus* [132] by directly inhibiting the autophosphorylation of FixL without affecting its dephosphorylation rate. In the closely related *Rhodobacter sphaeroides*, however, Osp, an SDRR protein, inhibits the kinase activity of CckA through a negative feedback loop. The inhibitory function of Osp and the similar action of the FixT protein suggest the existence of a group of RR-like proteins whose main function is to interact with the HK and prevent its phosphorylation [130]. The inhibition of HKs by single-domain RRs could be a common mechanism and can be implicated in regulating HKs that are structurally more complex than canonical HKs.

Proteolysis represents a rapid, irreversible means to silence TCS signaling. Several ATP-dependent proteases (such as ClpXP, Lon, and FtsH) are known to target RRs and sometimes HKs for degradation. Additionally, in some cases (such as in DegU~P), only one form of the protein—either phosphorylated or unphosphorylated—is preferentially degraded, adding an extra layer of regulatory control [133]. Interestingly, the protease FtsH was shown to protect RR PhoP from ClpAP proteolysis [134,135]. Additionally, regulatory sRNAs can influence TCS protein levels by affecting translation (e.g., MicA and GcvB bind the *phoP* mRNA and repress translation) [136,137]. These examples highlight the diversity of negative regulation of TCS by controlling protein levels.

Proper silencing mechanisms enable rapid adaptation to changing environments and allow for a swift return to a baseline once a stimulus is removed. Avoiding unnecessary gene expression helps bacteria optimize resource use, which is critical for survival under fluctuating conditions. The balance between positive and negative controls prevents inappropriate cross-talk between TCS pathways and ultimately shapes the output response [138]. Overall, the negative control fine-tunes bacterial signaling, enabling efficient adaptation and survival in diverse environments.

**Table 5 microorganisms-13-02013-t005:** Examples of the negative regulation of TCSs.

Target TCS Components	Organism	Biological Functions	Mechanism of Negative Regulation	References
CheY~P (HK)	*E. coli*	Chemotaxis	Spontaneous autodephosphorylation and deactivation	[107]
NtrB (HK)	*E. coli*	Nitrogen assimilation regulation	Increased phosphatase activity of NtrB after binding to the PII protein	[128]
KinA (HK)	*B. subtilis*	Inhibition of sporulation under unfavorable environmental conditions	Inhibition of KinA by SdA and KipI (antikinase); regulated by KipA	[125,126]
FixL (HK)	*R. meliloti*/*C. crescentus*	Repression of nitrogen fixation and the heme biosynthesis pathway	FixL autophosphorylation directly inhibited by FixT (SDRR)	[132,139]
CckA (HK)	*R. sphaeroides*	Regulation of cell cycle progression	Inhibition of CckA by Osp (SDRR)	[130]
BphP (HK)	*D. radiodurans*	Resistance to ionizing radiation	Red light activation of phosphatase activity	[123,124]
CopS (HK)	*P. aeruginosa*	Regulation of free copper ions in the periplasmic area	CopS phosphatase activity regulated by Cu^+^ (active in the absence of Cu^+^)	[102]
PhoQ (HK)	*Salmonella enterica*/*E. coli*	Regulation of the virulence-related system (*phoPQ*)	PhoQ phosphatase activity induced via a direct interaction with MgrB (kinase activity is inhibited)	[127]
EnvZ (HK)	*E. coli*	Part of the envelope stress response	Modulation of EnvZ activity by MzrA	[129]
Spo0A~P (RR)	*B. subtilis*	Inhibition of sporulation initiation	Specific dephosphorylation of Spo0A~P by Spo0E	[131]
Spo0F~P (phosphotransfer RR)	*B. subtilis*	Prevents sporulation during competence	Dephosphorylation of Spo0F~P by RapE	[131]
DegU~P (RR)	*B. subtilis*	Motility; biofilm formation; competence	Preferential degradation of the phosphorylated form	[133]
PhoP (RR)	*E. coli*	Maintenance of the envelope stress response and virulence regulation	ClpAP-mediated proteolysis, prevented by FtsH	[134,135]
phoP mRNA	*E. coli*	Posttranscriptional regulation of the virulence-related system (phoPQ)	Translational repression of the phoP mRNA by *micA* and *gcvB* sRNA bind	[136]

## 10. Cross-Talk Between TCSs

TCS cross-talk is defined as phosphotransfer from the HK of one TCS to the RR of another TCS (Figure 5). Consequently, the absence of one of these TCSs may not influence the expression of the regulated genes when the other TCSs are active [140]. Typically, efficient cross-talk in nature is rare—the observed noncognate interactions are rather the “side effect” of their common evolutionary origin. Bacteria typically acquire new TCSs through gene duplication, which initially allows for cross-talk before these duplicated TCSs diversify into distinct pathways. Interestingly, as few as two amino acid mutations can abolish the cross-talk between TCSs, underscoring the evolutionary pressure favoring specificity [141].

In *E. coli*, approximately 3% of the interactions between TCS proteins involve noncognate HK–RR pairs. Conversely, in *M. tuberculosis*, a significantly larger fraction—50% of interactions—occurred between noncognate HK–RR pairs [141]. Most information on this phenomenon comes from experiments performed in vivo on deletion mutants of some TCS components (e.g., deletions of cognate sensors) or in vitro in an isolated environment [142]. A study conducted in *E. coli* and *C. crescentus* showed that phosphotransfer between cognate HK–RR pairs is 1000-fold faster than that between noncognate pairs [143]. Interestingly, of the five systems studied in *M. tuberculosis*, unphosphorylated HKs showed at least 2-fold higher affinity for noncognate RRs (sometimes even 40-fold higher) than cognate partners did, potentially creating a mechanism to prevent weak or unintended responses [49]. The phenomenon of TCS cross-talk is considered detrimental because it disperses the signal, reducing the levels of the cognate RR-P and thereby weakening the response [141]. Kinetic studies have proposed that HK phosphatase activity functions to limit cross-talk between highly homologous TCSs [144].

Cross-talk that preconditions bacteria for upcoming signals may actually enhance adaptive responses and confer evolutionary advantages (Table 6). *E. coli* PhoB can be activated by PhoR and CreC, connecting phosphate metabolism with carbon catabolism [145,146], whereas OmpR can be phosphorylated not only by its cognate sensor, EnvZ, but also by ArcB, thus integrating anaerobic respiratory signals into porin regulation in *E. coli* anaerobiosis (Figure 5A) [147]. RcsB from *Salmonella typhimurium* was shown to undergo condition-specific phosphorylation not only by the cognate phosphorelay kinase but also by the noncognate phosphorelay sensor kinase BarA (Figure 5A) [148]. In *Enterococcus faecium*, efficient cross-talk between VanS (the TCS responsible for vancomycin resistance) and PhoB was detected (Figure 5B); however, the rate of transfer was at least 100-fold slower than that observed between the cognate pair of phospho-VanS and VanR [149]. It should be expected that many more other examples of cross-talk remain to be identified.

The specificity of TCSs is sometimes reduced for the efficient integration of signals from different sources; then, multiple TCS sensors interact with one RR (Table 6, Figure 5C). Such signal integration requires kinases that sense different signals to exhibit homology within the phosphotransfer domain. This situation is observed in the convergence of sensor proteins that control sporulation in *B. subtilis*, where five HKs (KinA–E) phosphorylate Spo0F [150]. Signal integration is also observed in quorum sensing in *Vibrio harveyi*: two different HKs, LuxN and LuxP, recognize different signal particles and phosphorylate the LuxU phosphotransferase protein, which eventually leads to the phosphorylation of one RR, LuxO [151,152]. In *E. coli* chemotaxis, the HK CheA, together with the coupling protein CheW, phosphorylates the RRs CheY and CheB (Figure 5D) [153]. A similar situation is observed for NarQ and NarX, which share homology within the sensor domain; both respond to nitrate and nitrite concentrations (although in different ways) and phosphorylate NarP and NarL, RRs that recognize distinct target sequences. Thus, they generate a cellular response that is adequate for the nitrate/nitrite ratio [154]. The activation of a regulator by multiple sensors broadens the spectrum of environments in which the RR regulon is expressed without evolving binding sites for different regulators at each of these genes. Altogether, these examples illustrate how cross-talk and signal integration within TCSs enhance regulatory flexibility, enabling bacteria to fine-tune gene expression in response to complex and fluctuating environments.

Based on the observations described above, mechanisms for RR phosphorylation known as many-to-one or one-to-many, where many HKs phosphorylate a given RR or a single HK phosphorylates multiple RRs, have been proposed. However, without mechanisms that control the specificity of communication between each HK and its cognate RR, unlimited cross-talk between TCSs would result in signal interference and would impede responses to activating stimuli. The identified mechanisms that uphold intrasystem signaling precision and curb intersystem cross-talk include molecular recognition between cognate HK and RR pairs, as well as the phosphatase activities exhibited by certain HKs, which regulate the phosphorylation levels of cognate RRs (as described above).

**Table 6 microorganisms-13-02013-t006:** Cross-talk between TCSs in bacteria. Noncognate interactions between HKs and RRs, where signal transduction is shared or redirected across systems, are listed.

HK/RR	Organism	Biological Functions	Type of Interaction	Mechanism	References
PhoR, CreC/PhoB	*E. coli*	Integrating phosphate metabolism and carbon catabolism	Many-to-one (cross-talk)	PhoB might be activated by cognate PhoR and noncognate CreC	[145,146]
EnvZ, AcrB/OmpR	*E. coli*	Regulation of porins under anaerobic and osmotic stress conditions	Many-to-one (cross-talk)	OmpR might be activated by cognate EnvZ and noncognate ArcB	[147]
RcsC, BarA/RcsB	*S. typhimurium*	Response to environmental changes and envelope stress	Many-to-one (cross-talk)	RcsB might be activated by the canonical activation pathway (phosphorelay) via RcsC (HK) and RcsD (phosphotransfer protein) or by noncognate BarA (HK)	[148]
VanS/PhoB, VanR	*E. faecium*	Signal transfer modulation during the antibiotic resistance response	One-to-many (cross-talk)	VanS might activate cognate VanR and noncognate PhoB	[149]
KinA–KinE/Spo0	*B. subtilis*	Integration of stress signals during sporulation initiation	Many-to-one (signal integration)	Five different kinases (KinA–KinE) might phosphorylate Spo0F	[150]
LuxN, LuxP/LuxU	*V. harveyi*	Integration of quorum sensing signals	Many-to-one (signal integration)	Two different kinases (LuxN and LuxP) recognize different signaling molecules and phosphorylate LuxU	[151,152]
CheA, CheW/CheY, CheB	*B. subtilis*	Integration of signals during chemotaxis	Many-to-many (signal integration)	Two HKs: CheA and CheW can phosphorylate two RRs: CheY and CheB	[153]
NarQ, NarX/NarP, NarL	*E. coli*	Coordinated regulation of nitrate/nitrite respiration	Many-to-many (signal integration)	NarQ and NarX can phosphorylate NarP and NarL in a nitrate and nitrite concentration-dependent manner	[154]

## 11. Hierarchical Actions and Cooperation of TCSs

In addition to cross-talk, in which different TCSs act in parallel, controlling the same targets, a few cases of hierarchical actions of TCSs have been reported where one TCS activates the transcription of another (Table 7, Figure 5E). For example, in response to an as yet undetermined signal, EnvZ-OmpR directly activates *ssrA*/*B* TCS transcription in *S. typhimurium*. Interestingly, as *ssrA* and *ssrB* transcription are not coupled, they are independently regulated by OmpR [56,155]. Similar hierarchical activation was observed in *S. coelicolor*, where the growth rate controlling SatKR activated the transcription of genes encoding the SitKR TCS [54]. In contrast, the *E. coli* phospho-NarL protein from the NarLX system inhibits the transcription of the *dcuSR* TCS, which indirectly leads to the repression of fumarate respiration when nitrate respiration is activated [14]. This finding shows that hierarchical action does not always lead to activation but can also lead to the inhibition of the transcription of another TCS. This sequential action of TCSs can also be indirect and involves other regulatory proteins, such as PhoP, which controls the phosphorylation state of PmrA in *Salmonella*. This system is also the first known example of a protein enabling one TCS to respond to the signal of another by protecting the phosphorylated form of an RR [156].

**Table 7 microorganisms-13-02013-t007:** Hierarchical cooperation of the TCSs for bacterial signaling networks. Examples of regulatory hierarchies where one TCS influences the expression or activity of another TCS or where multiple TCSs control the same target genes are listed.

Primary TCS/Secondary TCS/Other Targets	Organism	Biological Functions	Mechanism	References
EnvZ-OmpR/SsrAB	*S. typhimurium*	Virulence, curli fimbria synthesis	Transcriptional activation: OmpR activates *ssrA* and *ssrB* independently	[56,155]
SatKR/SitKR	*S. coelicolor*	Growth rate regulation	Transcriptional activation: SatKR activates *sitKR* genes	[54]
NarLX/DcuSR	*E. coli*	Coordination of fumarate and nitrate respiration	Transcriptional repression: NarL~P blocks the transcription of *dcuSR*	[14]
PhoPQ/PmrAB	*Salmonella*	Resistance to antimicrobial peptides	PhoP activates PmrD, which protects PmrA~P from dephosphorylation	[157]
PhoPR/ResDE	*B. subtilis*	Integrates phosphate signals into ATP production	Transcriptional activation: PhoPR activates *resDE*	[156]
PhoPR, ResDE, Spo0AB/Pho regulon	*B. subtilis*	The response to phosphate deficiency is correlated with the cell cycle	Coregulation of the same genes	[158]
CusRS, CopRS, CzcRS/copper resistance genes	*P. aeruginosa*	Zinc and copper homeostasis	Coregulation of the same genes	[159]
PhcA, PhcB, XpsR, VsrA-VsrD, and VsrB-VsrC/Eps (viral factor)	*R. solanacearum*	Virulence regulation by multiple signals	Coregulation of the same genes	[160]
PmrAB, PhoPQ, RcsCB/Ugd (lipopolysaccharides modifying enzyme)	*Salmonella*	Regulation of antibiotic resistance by multiple environmental signals	Coregulation of the same genes	[161]

An intrinsic property of TCSs is that the transcription of TCS-controlled genes requires both the RR and the signal detected by the cognate HK. Therefore, if one TCS controls the transcription of another, the genes regulated by the latter system will be expressed only when signals activating both systems are present. An example is the link between the PhoP/PhoR and ResD/ResE systems [157]. The transcription of the ResD/ResE system, which regulates anaerobic respiration genes in *B. subtilis* (in response to oxygen limitation or redox changes), is controlled by the PhoP/PhoR system, which responds to phosphate starvation. Low phosphate levels activate both the PhoP and ResD regulators, increasing the expression of *resDE* genes and creating a positive feedback loop that enhances PhoP/PhoR activation. This loop integrates phosphate signals into ATP production [157]. Transcriptional studies in *E. coli* and *B. subtilis* have shown that this type of intersystem control, which requires two signals, is common [162,163].

The expression of some genes can be controlled by multiple TCSs (Figure 5F). This process is due to the presence of multiple regulatory protein binding sites in their promoters. The regulation of a set of genes by two or more different TCSs, together with other regulatory proteins, leads to interplay between regulatory proteins without direct interactions. The best described phenomenon of this type is the regulation of a set of genes involved in the control of phosphate metabolism. However, the mechanism by which phosphate deficiency signals the RR is still unclear and probably varies among bacteria. Phosphate metabolism genes in *B. subtilis* are under the control of at least three TCSs, forming a regulatory network that mediates the response to phosphate deficiency. In addition to the abovementioned positive feedback between PhoPR and ResDE, the interconnected pathways involve the Spo0AB phosphorelay system, making this regulation connected to cell cycle control and thus even more complex [158]. In *E. coli*, the phosphate-sensing pathway, in addition to the PhoR–PhoB TCS, requires as many as five additional proteins [164], setting it as an excellent example of complex control.

Similar coregulation to that of phosphate metabolism was observed in the control of the intracellular copper level in *P. aeruginosa*. In response to copper in the environment, the CusRS TCS activates its own genes and an operon, providing resistance to high copper levels. This locus is also regulated by another copper-responsive TCS, CopRS, as well as zinc-responsive CzcRS TCSs. All three systems control the same genes but yield different activation levels. Surprisingly, cross-talk between CusRS and CopRS was observed even when one of the HKs was absent, suggesting interactions between these independent systems [159]. Similar coregulation and signal integration were shown for the expression of the Eps virulence factor in *Ralstonia solanacearum* [160] and the *Salmonella ugd* gene encoding an enzyme required for resistance to the antibiotic polymyxin B [161]. This complex regulation allows bacteria to respond to multiple environmental cues.

## 12. Phosphorelay

Although the general model of TCSs is relatively simple, these systems can in fact be part of complex phosphorelays involving multiple proteins, including sensory HKs, RRs with an effector domain, additional histidine phosphotransfer proteins, and SDRRs (Figure 6, Table 8). Both phosphoamino acids involved in phosphorylation mechanisms, phosphoHis and phosphoAsp, possess a significant free energy of hydrolysis. Therefore, phosphoAsp in the RR can initiate a reverse transfer of phosphate to phosphoHis in another His-containing phosphotransfer domain (HPt) [25,165,166]. In this arrangement, a sensor HK initiates the transfer of a phosphoryl group to an RR that contains the conserved aspartate domain but lacks an output domain. The RR then transfers the phosphoryl group to a phosphotransfer protein that contains a histidine. This protein, in turn, acts as the source of phosphorylation for the ultimate RR. This RR possesses an output domain that facilitates a cellular response. In some phosphorelay pathways, the HK and the RR with no output domain (and sometimes the His-containing phosphotransfer protein) are fused in a single polypeptide [167] (Figure 6). The transfer of the phosphoryl group from histidine (His) to aspartic acid (Asp) remains the same, regardless of the complexity of the system and the number of components involved [166]. As a result, the TCS signal transduction pathway can be involved not only in two-step phosphotransfer (His–Asp, Figure 2) but also in a multistep phosphorelay cascade (His–Asp–His–Asp, Figure 6) [166].

The presence of multiple stages in a phosphorelay makes this type of TCS an ideal solution for bacteria with a complex life cycle, as it offers additional potential targets for precise control of TCS activity, necessitating the efficient supervision of cell development stages [168]. An example of a bacterium employing such a complex phosphorelay is *B. subtilis*, whose sporulation is triggered by suboptimal conditions. Signal transfer between proteins must be dynamic to efficiently initiate sporulation. A critical point in the initiation of sporulation in *B. subtilis* is the activation of the main transcriptional regulator Spo0A. Its phosphorylation and, at the same time, activation occur by the multicomponent pathway involving the phosphorylation of the two RRs Spo0F and Spo0A [169]) and autophosphorylation of HKs (KinA-KinE) [170].

Another well-described example of a complex phosphorelay is that in Enterobacteriaceae, which consists of three proteins: RcsC, RcsD, and RcsB. In *E. coli*, the Rcs phosphorelay is responsible for biofilm formation, but in pathogenic bacteria such as *S. enterica* and *Proteus mirabilis*, it controls swarming motility [171]. The Rcs phosphorelay system also requires many accessory proteins to function properly, such as the lipoprotein RcsF (required for the reception and accumulation of signals that activate the Rsc phosphorelay) and the IgaA protein (a cytoplasmic membrane protein that inhibits the activity of RcsC kinase) [171]. Other examples of phosphorelay pathways are shown in Table 8.

**Table 8 microorganisms-13-02013-t008:** Representative bacterial phosphorelay systems, including classical multicomponent and hybrid HK-based TCSs. The table outlines the types of signaling pathways, biological functions, and unique mechanistic features.

HK/RR	Organism	Biological Functions	Type of Regulation	Mechanism	References
KinA-KinE/Spo0A, Spo0F	*B. subtilis*	Sporulation initiation	Multicomponent phosphorelay	Involves multiple regulators and phosphatases for signal integration and control	[131,169,170]
RcsC, RcsD (HPt)/RcsB	*E. coli*	Biofilm formation, motility control	Three-component phosphorelay	Involves accessory proteins like RcsF and IgaA	[171]
ArcB (hybrid HK)/ArcA	*E. coli*	Anaerobic respiration regulation	Hybrid HK phosphorelay	Involves cytoplasmic HK, responsive to the redox state	[172,173]
CckA (hybrid HK), ChpT (HPt)/CtrA	*C. crescentus*	Cell cycle regulation, DNA replication block	Hybrid HK phosphorelay	Phosphotransfer from CckA to ChpT; controls the master regulator CtrA	[174]
VirA (hybrid HK)/VirG	*Agrobacterium tumefaciens*	Plant infection (virulence gene expression)	Hybrid HK phosphorelay	Phosphorelay via an internal receiver and HPt domains in VirA	[175]

## 13. Hybrid Sensor Kinases in Phosphorelays

Phosphorelay systems involve multiple phosphorylation steps, and in many bacteria and some archaea, these pathways are mediated by hybrid HKs. Hybrid HKs are characterized as proteins with key domains involved in phosphotransfer, i.e., the kinase domain and REC domain and, sometimes, an HPt domain (like in VirA from *A. tumefaciens*) located in one polypeptide [176] (Figure 7). Previously, hybrid HKs were reported only in eukaryotic organisms [167]. Recently, hybrid-type HKs have also been detected in almost half of the sequenced prokaryotic genomes. The first hybrid HK in prokaryotes was identified in *E. coli* as the product of the *arcB* gene [172]. Compared with traditional HKs, hybrid HKs are significantly more prevalent in Gram-negative bacteria, especially in lineages with high regulatory demands (e.g., Proteobacteria), enhancing the diversity and adaptability of two-component signaling networks [25,177].

Hybrid HKs are reportedly derived from the lateral transfer and duplication of the REC domain [178]. When genes encoding an HK and RR colocalize near each other, the likelihood of hybrid protein formation increases. The appearance of the hybrid gene is suspected to result from mutations in the stop codon and read-through from the HK gene [179,180]. Such a fusion of an HK and RR into one protein demonstrates the advantage of a single-component system where the entire signaling process takes place in one protein [176,179]. In addition to the kinase and REC domains, the hybrid HK can also have an HPt domain, resulting in phosphotransfer within the same polypeptide.

However, many prokaryotic hybrid HKs do not transfer phosphate directly within their own domains but instead rely on a separate HPt protein. For example, in *C. crescentus*, the hybrid HK CckA passes its phosphate to the HPt protein ChpT, which then relays the signal further [75]. Using a standalone HPt protein allows more flexible regulation because a single HPt (such as ChpT) can integrate signals from several different HKs.

Some hybrid HKs lack an internal HPt domain altogether and instead “borrow” one from another protein, as observed in *Coxiella burnetii*, *M. xanthus*, and *Pseudomonas* spp. [25,165,181,182,183]. In other cases, hybrid HKs contain extra sensory modules—such as CheW, GAF, or PAS domains—like those found in *Synechocystis* sp. These additional domains help the kinase detect specific environmental cues and adjust signaling accordingly [184].

Although the exact structures differ across species, the design of hybrid HKs is usually fine-tuned for particular signaling needs. Rather than simply adding more inputs, these extra domains often refine the specificity of the kinase for its regulator [185] or modulate its autokinase and phosphotransfer activity [186,187].

In summary, hybrid HKs demonstrate how TCSs can diversify their architecture: by using separate HPt proteins or adding extra sensory domains, these systems achieve greater regulatory flexibility, signal integration, and specificity for their RRs. This modular design highlights the evolutionary strategies bacteria use to fine-tune signaling networks.

## 14. Negative Control of Phosphorelays

Phosphorelays incorporate multiple regulatory checkpoints (provided by many internal domains, e.g., REC and HPt)**,** allowing tunable signal shutdown, flexibility, and temporal control. However, they also require more sophisticated mechanisms to ensure signaling fidelity (such as dephosphorylation). The multistep nature of phosphorelays permits inhibitory control at each phosphotransfer point. Each phosphoacceptor—REC or HPt—can be a target for dephosphorylation either through intrinsic hydrolysis or through regulated enzymatic activity. For example, in the ArcB–ArcA system of *E. coli*, ArcB contains a complete phosphorelay cassette (His–Asp–His), and its internal REC and HPt domains participate in both signal propagation and reversal through dephosphorylation [173,188,189]. In phosphorelays, similar to classical TCS sensors, HKs may be bifunctional, i.e., act as either kinases or phosphatases depending on the environmental conditions or interacting partners. In phosphorelay systems, hybrid HKs frequently exhibit autophosphatase activity, catalyzing the removal of phosphoryl groups from their internal REC domains or from downstream RRs. For example, the hybrid HK CckA, which controls the phosphorylation state of the master regulator CtrA in *C. crescentus*, toggles between kinase and phosphatase modes during the cell cycle, depending on the protein interactions (e.g., with DivL or c-di-GMP levels) [190,191]. The advantage of a phosphorelay is the possibility of more complex and multistep silencing of unwanted responses to weak or short stimuli by the deactivation of the HK and inhibition of phosphotransfer.

Some phosphorelays are modulated by dedicated phosphatases that specifically target phosphorylated HPt domains or terminal RRs. A well-characterized example is SixA, an auxiliary protein in *E. coli* that accelerates dephosphorylation of the HPt domain in ArcB, thereby terminating ArcA activation and repressing the downstream transcriptional response [172]. These auxiliary phosphatases act independently of the core phosphorelay proteins and may serve as checkpoints linking the metabolic status to the signal output. Additionally, the phosphorylated state of an RR can be controlled, for example, by aspartyl phosphate phosphatases, allowing the integration of multiple signals through a His-Asp-His-Asp phosphorelay system.

In the *B. subtilis* phosphorelay pathway, sporulation is controlled by several phosphatases that dephosphorylate the Spo0A and Spo0F RRs. Spo0A~P is specifically and directly modulated by the Spo0E phosphatase in response to signals that remain unknown. Spo0F∼P is the known target of RapE, which is specifically controlled by another TCS, ComAP, and governs the development of competence, ensuring that sporulation and competence do not develop at the same time [131]. In contrast, the transcription of two other related phosphatases controlling Spo0F, RapA and RapB, is induced by growth conditions that maintain the cells in a vegetative state [192]. Thus, division and other cellular processes (e.g., competence) that are not compatible with sporulation prevent its initiation by the specific dephosphorylation of the phosphorelay.

In some systems, accessory proteins serve as scaffolds or inhibitors that modulate HK activity. In the Rcs phosphorelay of *S. enterica*, the inner membrane protein IgaA suppresses the autokinase activity of the hybrid HK RcsC under nonstress conditions [193]. Upon envelope stress, IgaA is inactivated, allowing the phosphorelay to proceed through RcsD (HPt) to the RR RcsB. This mode of repression ensures that Rcs signaling remains silent in the absence of appropriate stimuli. In addition to enzymatic dephosphorylation, negative control of phosphorelays can also be achieved through conformational changes that restrict domain accessibility. In hybrid HKs, the interaction between the CA domain and the REC or HPt domains may be sterically hindered or allosterically regulated by internal or external signals, affecting phosphotransfer efficiency. This structural gating serves as an additional regulatory layer, allowing the rapid, reversible modulation of signaling flux without altering protein abundance or localization.

## 15. Summary

TCSs are widespread in bacteria and play a crucial role in adaptation to changing environments, enabling regulation of diverse aspects of bacterial life cycles. Despite decades of intensive study, TCSs continue to reveal unexpected complexity. Key knowledge gaps remain regarding how bacteria achieve signaling specificity among dozens of parallel pathways, the physiological relevance of cross-talk and hierarchical interactions, and the influence of non-canonical phosphorylation events, tandem or hybrid HKs, accessory regulators, or atypical domain architectures on regulatory outcomes.

Emerging technologies are beginning to address these challenges. High-resolution cryo-EM and integrative structural biology are uncovering new conformational states and allosteric mechanisms [47,194]. Phosphoproteomics and single-cell omics provide systems-level insights into TCS signaling networks and their plasticity in fluctuating environments [195]. CRISPR-based genome engineering and synthetic biology approaches now allow targeted rewiring of TCS circuits, offering both functional insight and opportunities for innovative applications [196]. These advances have already revealed unexpected cross-regulatory interactions and context-dependent signal integration, highlighting a greater diversity of regulatory logic than previously appreciated.

Beyond advancing fundamental understanding, these insights point to opportunities for novel antimicrobial strategies, microbiome interventions, and biotechnological tools. Synthetic biologists are repurposing TCSs for applications in optogenetics, materials science, gut microbiome engineering, and soil nutrient sensing. Engineered TCS modules are being used to detect heavy metals and disease-related biomarkers, as well as to produce valuable bioproducts [196]. New engineering strategies, such as genetic refactoring, swapping DNA-binding domains, adjusting detection thresholds, and preventing phosphorylation cross-talk, are improving TCSs’ sensor performance and making them more reliable for specific applications [30,31,123,196,197,198,199]. When combined with large-scale gene synthesis and laboratory-based screening, these approaches could facilitate the discovery of the stimuli detected by many uncharacterized TCSs, potentially yielding a broad family of genetically engineered sensors capable of detecting diverse inputs [200]. However, many TCS-based biosensors still face performance limitations due to the lack of effective engineering strategies [31].

Because TCSs are common in bacteria but absent in animals, they are increasingly recognized as attractive antimicrobial targets, given their central role in virulence, stress resistance, and biofilm formation [201]. One promising approach involves developing inhibitors of RRs that either possess antibacterial activity or enhance the effects of existing antibiotics, a strategy that has not yet been extensively reviewed [202,203]. Furthermore, selective modulation of TCS activity could provide strategies to reprogram microbiomes or attenuate pathogens without eradicating them, offering alternatives to conventional antibiotics [204,205,206,207].

Taken together, we anticipate that the coming years will see rapid progress in dissecting the fundamental biology of TCSs while translating this knowledge into practical applications. By integrating molecular detail with systems-level understanding, future research will illuminate how bacteria adapt with such versatility and exploit this versatility for biotechnology and medicine. Nevertheless, the vast complexity of TCSs, including their cross-talk, dependencies, and connections to other regulatory systems, continues to pose significant challenges for their controlled rewiring and application. In particular, non-typical TCSs—such as hybrid kinases, regulators lacking canonical domains, and systems that employ auxiliary proteins or multistep phosphorelays—remain especially challenging to study. Their complex architectures, context-dependent signaling, and integration with other pathways complicate functional annotation and mechanistic understanding, but at the same time provide valuable insights into the evolutionary flexibility and regulatory diversity of bacterial signaling networks.

## Figures and Tables

**Figure 1 microorganisms-13-02013-f001:**
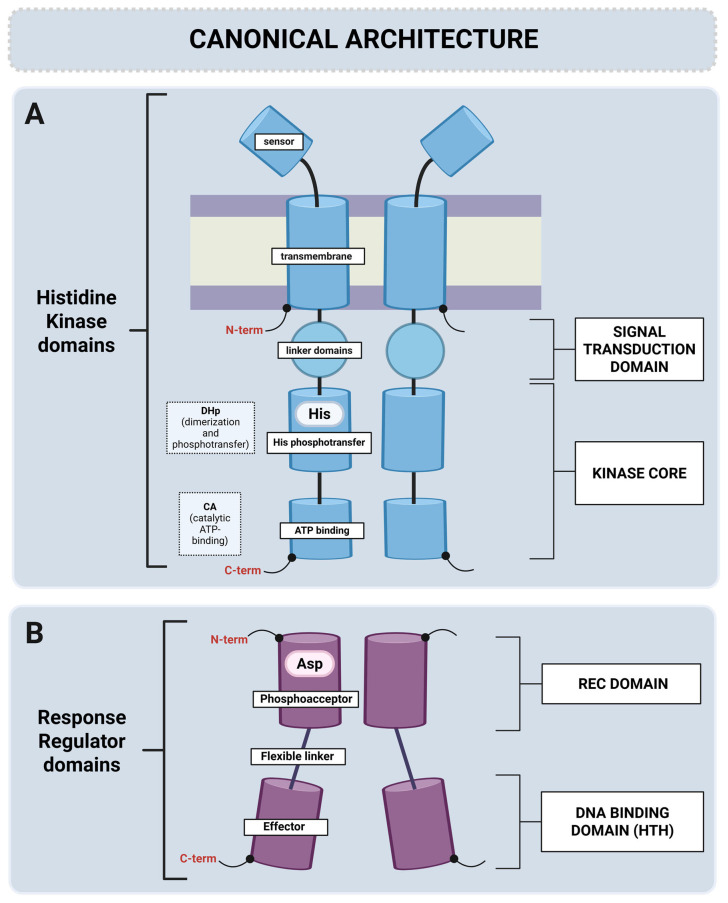
Architecture of a canonical TCS. (**A**) Typical domain structure of an HK encompassing a sensor domain, transmembrane domain, linker domains (e.g., HAMP, PAS, and GAF), and a conserved kinase core domain composed of DHp (with the histidine residue being phosphorylated) and CA (ATP-binding) domains. (**B**) The structure of a canonical RR consists of an REC domain with a conserved aspartate linked by a flexible region to an effector DNA-binding domain. Notably, RR variations exist, including non-DNA-binding or phosphorylation-independent regulators. Created in BioRender. https://BioRender.com/ywgajc7 (accessed on 25 July 2025).

**Figure 2 microorganisms-13-02013-f002:**
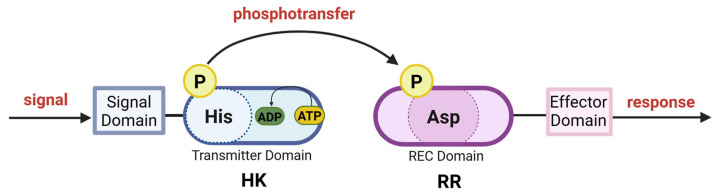
Canonical signal transduction in a TCS. Upon sensing a signal, an HK undergoes autophosphorylation at a conserved histidine residue (His) and transfers the phosphate group (P) to an aspartate residue (Asp) on an RR, which then activates its output domain to trigger a cellular response. Created in BioRender. https://BioRender.com/50id6qd (accessed on 25 July 2025).

**Figure 3 microorganisms-13-02013-f003:**
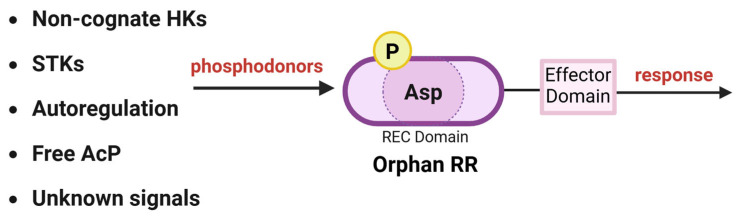
Possible activation mechanisms of orphan RRs. Orphan RRs, which lack a known cognate HK, can be activated by alternative phosphodonors, such as noncognate HKs, protein serine/threonine kinases (STKs), and AcP, or through autoregulation and unknown signals. Created in BioRender. https://BioRender.com/2okfvtj (accessed on 25 July 2025).

**Figure 4 microorganisms-13-02013-f004:**
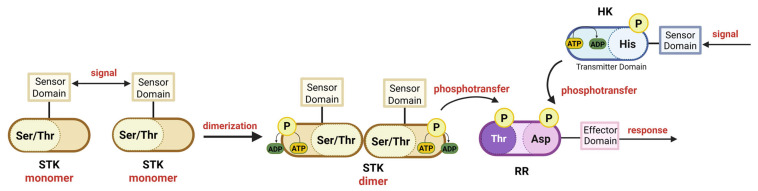
Activation of RRs by STKs. Upon sensing a signal, STK monomers dimerize and autophosphorylate a serine/threonine residue using ATP. The phosphate is then transferred to a threonine residue in an RR, triggering the activation of the output domain and a downstream response. Specific examples are listed in Table 4. Created in BioRender. https://BioRender.com/35yhdjf (accessed on 25 July 2025). Specific examples are listed in Table 4.

**Figure 5 microorganisms-13-02013-f005:**
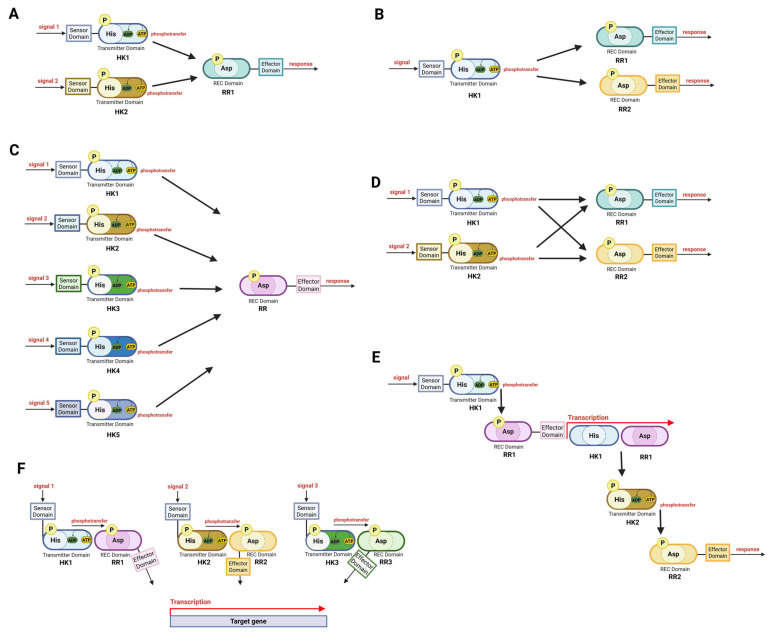
Cross-talk and cooperation of TCSs. The schematics illustrate non-cognate interactions. (**A**) Cross-talk “many-to-one,” where two distinct HKs (HK1 and HK2) phosphorylate a single RR (RR1). (**B**) Cross-talk “one-to-many,” where one HK (HK1) phosphorylates two distinct RRs (its cognate RR1 and a non-cognate RR2). (**C**) Signal integration “many-to-one,” where multiple HKs (HK1–HK5), responding to diverse signals, phosphorylate and transfer the signal to one RR. (**D**) Signal integration “many-to-many,” where two RRs (RR1 and RR2) are simultaneously phosphorylated by two HKs in response to two different signals. (**E**) Hierarchical action of TCSs, where TCS2 is transcribed in response to a signal transmitted by another TCS (TCS1). (**F**) The expression of target genes controlled by multiple TCSs. Created in BioRender. https://BioRender.com/9s65rhu (accessed on 25 July 2025). Specific examples are listed in Table 6 and Table 7.

**Figure 6 microorganisms-13-02013-f006:**
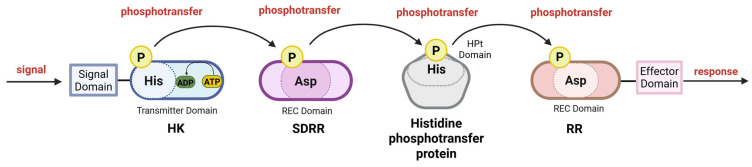
Classical four-component phosphorelay system in bacteria. The multistep phosphorelay cascade where a canonical HK first autophosphorylates and transfers the phosphate group to a SDRR is shown. The phosphoryl group is then relayed to a histidine phosphotransfer protein and finally to the terminal RR equipped with an output domain, triggering the cellular response. Created in BioRender. https://BioRender.com/u1becki (accessed on 25 July 2025). Specific examples are listed in Table 8.

**Figure 7 microorganisms-13-02013-f007:**
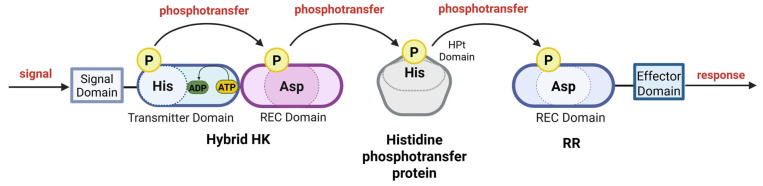
Hybrid HKs in the phosphorelay system. A hybrid HK first autophosphorylates and transfers the phosphate to its own REC domain. The signal is then relayed via an HPt protein to the final RR, which triggers the output response. Created in BioRender. https://BioRender.com/h2wo4og (accessed on 25 July 2025).

## Data Availability

No new data were created or analyzed in this study.

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
