# Peer review of "Untangling the Complexity of Two-Component Signal Transduction in Bacteria"

_microorganisms, 2025, doi:10.3390/microorganisms13092013_

Round 1

Reviewer 1 Report

Comments and Suggestions for Authors

I am surprised that the majority of sources in this work are from past decades. The primary purpose of a review is to acquaint readers with current perspectives. Is it possible that no significant studies have emerged in this field in recent years? For example, it was discovered in 2025 that DesK can mantain DesR activity without phosphorylation at all, just by protein-protein interactions (10.1038/s41598-025-88468-5). Also it was discovered that KdpD  acts as a tandem serine histidine kinase (10.1038/s41467-024-47526-8). I am not encouraging the inclusion of these specific works, but rather emphasizing the necessity to thoroughly examine recent studies that have not yet been cited in prior reviews.

“was first characterized in Escherichia coli and later characterized in many other bacterial species (Santos-Beneit et al. 2008).”
The manuscript uses two inconsistent citation styles and features inconsistent bold formatting (particularly at the start of sections 13 and 14, which gives an impression reminiscent of AI-generated text). Italicization is also occasionally missing (e.g., for Salmonella in section 4). The entire text requires careful manual verification.

 “Despite many HKs being membrane associated, they might respond to signals within the cytoplasm, such as changes in osmolarity, membrane fluidity (linked directly to temperature changes), and ion concentrations”
I would challenge these formulations. Membrane fluidity is a signal originating from the membrane itself, not the cytoplasm (I would clearly distinguish these two entities). Sensitivity to osmolarity could also reflect membrane stretching (as seen in PhoQ, for instance).

“PAS is a compact α/β fold that often binds cofactors such as FAD or heme, which detects small molecules, oxygen, redox potential, light, or other environmental stimuli, whereas GAF, which is structurally similar to the PAS domain but different in sequence, usually binds cyclic nucleotides, e.g., cGMP or cAMP, or other small ligands”
Please elaborate on how this affects conformational signal transmission within histidine kinases.

“While most HKs are located within the membrane, soluble HKs are also regulated by intracellular signals or through interactions with the cytoplasmic domains of other proteins [26–28]. For example, B. subtilis DesK HK lacks extracellular domains and instead relies on changes in the membrane thickness or composition to sense temperature fluctuations, effectively linking cytoplasmic conditions to cellular responses”
DesK is not a soluble HK; it is a transmembrane sensor.

“This process occurs through a bimolecular reaction between homodimers, where one HK monomer catalyzes the phosphorylation of the conserved histidine residue on the other monomer”
Both cis-phosphorylation and trans-phosphorylation are widespread (doi: 10.1016/j.jmb.2013.01.011,  10.1038/s42004-024-01272-6).

“How-ever, if the stimulus is weak or short-lived, this autoregulation might cause an exaggerated and prolonged response compared with the actual stimulus. Autoregulation usually allows the system to adjust gene expression finely, enabling different levels of the response regulator to control distinct sets of genes.”
Do these two sentences contradict each other? Clarify whether "exaggerated" here implies "inappropriate". This paragraph lacks citations supporting its claims.

“AcP has been shown to be the sole in vivo phosphodonor for a few RRs that appear to lack cognate HKs, including NtrC-like RR Rrp2 from Borrelia burgdorferi”
Data exists that contradicts this assertion (10.1371/journal.pone.0144472, 10.1128/jb.01010-15).

 “Additionally, regulatory sRNAs can influence TCS protein levels by affecting translation (e.g., MicA and GcvB bind the phoP mRNA and repress translation) [116].”
Reference 116 doesn’t mention GcvB.

“especially in lineages with high regulatory demands (e.g., Proteobacteria), where they account for 20–30% of all histidine kinases [37, 157].”
Could the specific source for this claim and the numerical estimate (20–30%) be clarified? The cited references [37, 157] do not appear to explicitly contain these numbers.

Reviewer 2 Report

Comments and Suggestions for Authors

The manuscript presents an extensive review of two-component systems (TCSs) in bacteria, covering everything from the canonical architecture to lesser-known structural, functional, and regulatory variations. However, it is not groundbreaking in the sense of presenting new experimental data, novel hypotheses, or original conceptual models.

Suggestions to strengthen the article:

Refine the Introduction – More clearly highlight the knowledge gap that motivated this review and justify its relevance compared to existing reviews in the literature.

Streamline the description of the canonical model (Section 2) – This section takes up considerable space and repeats widely known information. A more concise version would allow for greater emphasis on non-typical variations, which are the central focus.

Increase the critical nature of the review – Although the text comprehensively compiles the literature, it is predominantly descriptive. Include more analytical discussion, highlighting limitations of current studies, controversies, and hypotheses that deserve further investigation.

More forward-looking conclusion – Include a final paragraph with knowledge gaps, future trends, and potential applications of TCSs in biotechnology and medicine, reinforcing the translational value of the review.

Standardization of nomenclature – Standardize the use of full names and abbreviations for proteins and systems throughout the text and tables, ensuring clarity.

Integration of figures with the text – Explore the content of the figures more analytically, connecting them directly to the points discussed.

Practical applicability – Expand the discussion on how knowledge of non-typical TCSs can be exploited as targets for antimicrobials, in microorganism engineering, or in the development of biosensors.

In the appendix, I have provided some comments as suggestions.

Overall, this could have a significant impact on the scientific community interested in molecular microbiology and bacterial signaling.

Round 2

Reviewer 1 Report

Comments and Suggestions for Authors

The authors have addressed all the comments by correcting the shortcomings. I believe the article can now be accepted.

Reviewer 2 Report

Comments and Suggestions for Authors

The manuscript presents an organized and comprehensive review of two-component systems (TCS) in bacteria, addressing not only the canonical model but also functional variations, regulatory mechanisms, and signal integration. Major improvements have been made to the article, making it technically sound and citing current references, aligning it with the scope of Microorganisms. I suggest accepting the article.

Comments on the Quality of English Language

The English writing is clear and understandable, allowing for adequate communication of the manuscript's central ideas.